# Behavioral Activation through Virtual Reality for Depression: A Single Case Experimental Design with Multiple Baselines

**DOI:** 10.3390/jcm11051262

**Published:** 2022-02-25

**Authors:** Desirée Colombo, Carlos Suso-Ribera, Isabel Ortigosa-Beltrán, Javier Fernández-Álvarez, Azucena García-Palacios, Cristina Botella

**Affiliations:** 1Department of Personality, Evaluation and Psychological Treatments, Instituto Polibienestar, University of Valencia, Avd Blasco Ibañez 21, 46010 Valencia, Spain; 2Department of Basic Psychology, Clinic and Psychobiology, Jaume I University, 12071 Castellón, Spain; susor@uji.es (C.S.-R.); fernanja@uji.es (J.F.-Á.); azucena@uji.es (A.G.-P.); botella@uji.es (C.B.); 3Department of Information and Communications Technologies, Pompeu Fabra University, 08002 Barcelona, Spain; isabel.ortigosa@upf.edu; 4CIBER Fisiopatología Obesidad y Nutrición (CIBERobn), Instituto Salud Carlos III, 28029 Madrid, Spain

**Keywords:** virtual reality, behavioral activation, depression, single case experimental design, multiple baselines

## Abstract

Behavioral activation (BA) is a structured psychotherapeutic approach for the treatment of major depressive disorder (MDD), which aims at increasing the engagement in activities that might bring enjoyment and meaning to patients’ lives. Although a growing body of evidence supports the effectiveness of BA, enhancing the motivation and activity level of depressed patients is often challenging. In the present study, we explored the effectiveness of a brief BA treatment supported by virtual reality (VR) to facilitate the visualization and anticipation of four pleasurable activities that we tried to re-introduce in the patients’ daily routine. To do so, we conducted a single-case experimental design with multiple baselines in a sample of patients with moderate to severe depressive symptoms. Three overlap analyses across participants and across behaviors were conducted to calculate the rate of improvement of each patient after the delivery of the intervention. Across the three overlap indices, the participants generally showed moderate-to-large improvements in the level of daily activity, as well as in the time spent planning and/or engaging in one or more activities scheduled during the intervention. Furthermore, most patients also reported a moderate-to-large reduction in daily depressive symptoms and improved mood. Overall, the promising results of the present study suggest that the proposed VR-based BA intervention might represent a valid approach to behaviorally activate depressed patients. The barriers and future lines of research of this innovative field are discussed.

## 1. Introduction

Major depressive disorder (MDD) is a debilitating condition affecting 4.4% of the general adult population [1]. MDD is characterized by the presence of mood disturbances, including reduced positive affect (PA), high negative affect (NA), and impaired ability to regulate positive and negative emotions [2]. As such, depression can affect different areas of the patients’ daily functioning, thus leading to significant distress and low quality of life [3,4].

Among other symptoms, depressed patients have been shown to experience loss of interest and pleasure in daily activities, which often results in the avoidance of pleasant situations and, therefore, in high social isolation. In this vein, negatively biased expectations towards the future have been suggested as a key feature of depressive future-oriented cognition [5]. Depressed patients tend to predict and overestimate the number and intensity of future negative experiences, while anticipating fewer positive events [6,7,8]. As suggested in previous studies [9], this bias contributes to a maladaptive vicious cycle prolonging NA and maintaining depressive symptoms. More precisely, this negative future-oriented disposition would reinforce the avoidance of pleasurable and rewarding activities and reduce the likelihood to engage in behaviors that might contribute to increase patients’ sense of accomplishment and mastery. As a result, depressed patients experience less opportunities to improve their mood, which further intensifies symptom severity [10].

So far, a wide range of evidence-based treatments have been developed to target MDD symptoms. Among others, behavioral activation (BA) is a structured psychotherapeutic intervention that aims to increase the patients’ engagement in adaptive daily activities while decreasing the engagement in behaviors that potentially maintain or prolong depressive symptoms [11]. The main goal of BA is to behaviorally activate patients by promoting the re-engagement in activities that might bring pleasure and satisfaction (i.e., reinforcement) to life, according to one’s personal values. With this goal in mind, patients are encouraged and supported to act from the “outside-in”: that is, to act according to a structured schedule, instead of a mood [10]. In this sense, BA is a highly customized therapy, in which the alliance between the therapist and the patient is essential to recognize behavioral patterns related to one’s depression, as well as to identify tailored rewarding activities to be scheduled and re-introduced into one’s daily routine [10].

Over the past decades, a growing body of evidence has supported the effectiveness of BA for the treatment of MDD [12], and studies have shown that BA is as effective as other well-established psychological interventions, such as cognitive therapy and cognitive and behavioral therapy [13,14,15]. Besides, BA has shown its efficacy to reduce symptoms of anxiety [16], as well as to promote well-being in non-clinical populations [16]. However, as suggested by Paul et al. (2020), finding the motivation to behaviorally activate a depressed person can be quite challenging due to the intrinsic nature of the disorder. In this direction, mental imagery has been shown to be a powerful tool to boost people’s motivation to engage in adaptive behaviors [17]. In their study, Renner and colleagues (2019) revealed that guiding participants through the imagination of pleasant activities enhanced their motivation and likelihood to engage in those situations. Although effective with healthy individuals, positive imagery may not be an easy task for patients suffering from MDD [18,19], who usually struggle to imagine and think about positive experiences occurring in their lives [6,7,8]. Consistently, the development of novel approaches to administer BA and, especially, to motivate and facilitate the anticipation of joyful experiences through imagination, represents an important challenge.

Virtual reality (VR) has been extensively used for the treatment of several psychological conditions [20,21,22] and, more recently, it has been implemented for the treatment of depression [23,24,25,26] and the improvement of emotion regulation skills [27]. Thanks to the immersiveness and the sense of presence [28,29], VR has been proved to be highly effective for emotional induction procedures [30]. Moreover, the possibilities offered by recent software and hardware advancements have made it possible to produce tailored experiences which are extremely close to reality [31]. Accordingly, the use of VR might represent an interesting tool in the field of BA interventions. First, VR could be used to provide a virtual spatial reference that facilitates the visualization of activities that once brought pleasure into the patients’ daily lives, which might be more challenging to achieve through mental imagery only. Second, the adoption of VR for the induction of positive emotions has been shown to be highly effective [32,33,34], which indicates that VR might foster the anticipated pleasure and reward of an activity and, in turn, increase one’s motivation. Accordingly, Chen et al. [35] tested the effectiveness of a VR-based protocol to expose depressed patients to positive scenes, showing significant decreased levels of anhedonia and depressive symptoms. Third, VR has been shown to enhance the vividness of mental simulations thanks to the feeling of “being there” and the possibility to experience the virtual environment from a first-person perspective [36]. Since the generation of more vivid mental simulations has been associated with a higher induction of positive emotions [37,38], the use of VR might represent a powerful tool to potentiate BA interventions and promote patients’ engagement in personally rewarding activities. In this direction, a recent study integrated the use of 360° videos into a broader BA protocol for depression to increase motivation towards pleasurable activities [39]. Preliminary results from a case study demonstrated the feasibility and acceptability of the intervention, which was associated with reduced depressive symptoms. However, VR was only implemented as an adjunctive tool to be used by the patient alone between therapy sessions. Furthermore, the behavioral outcomes associated with the intervention were not analyzed, so the utility of VR to promote BA in depressed patients is still unclear.

The aim of this study was to test the effectiveness of a four-session, VR-assisted BA protocol for individuals with moderate-to-severe depressive symptoms. The novelty of this brief intervention relies on the use of VR to provide patients with a virtual spatial reference in which to visualize and anticipate personally pleasurable and rewarding activities to be reintroduced in their daily routines. Consistent with the ample evidence showing the effectiveness of BA to activate depressed patients [12,13,14,15], we hypothesized that our VR-based protocol would enhance patients’ behavioral activation and increase daily activity levels. More specifically, we explored whether the motivation and the time spent planning or practicing the activities included in the protocol improved after the administration of the intervention, as well as whether the overall level of patients’ daily activity and associated savoring increased over time. Consistent with the anticipated behavioral outcomes and with the evidence from the previous literature regarding BA efficacy [12], we also expected to observe improved daily mood and decreased depressive symptoms. 

## 2. Materials and Methods

### 2.1. Study Design

In the present study, we conducted a single-case multiple baseline experimental design [40], a type of single case experimental design (SCED) in which participants are randomly assigned to different starts of the intervention phase to guarantee the internal validity of the findings. The study was conducted in accordance with the Single-Case Reporting Guidelines in Behavioral Interventions (SCRIBE) checklist for SCEDs [41].

Consistent with the guidelines [42], three different baseline lengths were adopted, each of which followed an AB design (A = baseline, B = treatment). Group 1 included 8 days of baseline (phase A) and 23 days of treatment monitoring (phase B); group 2 included 9 days of baseline and 22 days of treatment monitoring; group 3 included 12 days of baseline and 19 days of treatment monitoring. The length of the baseline and follow-up periods was chosen in order to ensure a minimum of 5 assessments per participant, as recommended for the analyses of SCEDs [43]. In total, the duration of the study was of 31 days. All participants received two intervention sessions per week over two weeks, thus resulting in a total of four sessions. During the whole study duration, participants were asked to complete a daily electronic diary sent through email using the Qualtrics platform. The daily assessment was sent at 8:30 p.m. Participants were instructed to complete each daily questionnaire on the day of receipt (i.e., not retrospectively). Participants who completed more than 30% assessments in a retrospective way were excluded from the study.

The present study was approved by the ethics committee of the Jaume I University (number: CD/103/2021). The study was registered in the clinicaltrials.gov (accessed on 14 January 2022) database (NCT05138744). Informed consent was obtained from all participants.

### 2.2. Inclusion and Exclusion Criteria

To be eligible for the study, participants had to meet the following criteria: being aged between 18 and 65 years, scoring more than 10 at the Patient Health Questionnaire-9 (PHQ-9) [44,45] (i.e., from moderate to severe depressive symptoms) and scoring less than 24 at the PA subscale of the Positive and Negative Affect Schedule (PANAS) [46,47] (i.e., a standard deviation below the mean of healthy individuals) [48].

Individuals who were already receiving a psychological treatment were excluded from this study. Furthermore, we excluded individuals suffering from a severe mental disorder as assessed with the Mini International Neuropsychiatric Interview Version 5.0.0 (MINI) [49], such as bipolar disorder, alcohol and/or substance dependence disorder or psychotic disorder.

### 2.3. Sample

The study was conducted at the Psychological Care Center of the Jaume I University (Spain). In total, eight participants meeting the inclusion criteria were recruited to take part in the study. One participant completed 45% of the daily assessments retrospectively and was therefore excluded from the analyses. The final sample included seven participants (six females and one male). According to the baseline PHQ-9 scores and existing cut-offs [45,50], three participants reported moderate depressive symptoms (i.e., a score between 10 and 14), three participants reported moderately severe symptoms (i.e., a score between 15 and 19), and only one participant reported severe depressive symptoms (i.e., a score over 20). More details about the sample are provided in Table 1.

All participants met the criteria for a MDD, as assessed with the MINI. Participant 4 and participant 7 were receiving a pharmacological treatment for depression at the time of the study. However, as the medication had not been introduced or changed over the last year, both participants were included in the study. 

### 2.4. Intervention

The protocol consisted of four VR-based BA sessions, which were delivered twice a week over two weeks. Each session lasted between 30 and 45 min. Before the beginning of the study, participants were asked to select four activities from a predetermined list during a face-to-face meeting (see Appendix B), which took place the week before the intervention phase. The instructions encouraged the participants to choose activities that they liked but they no longer performed, as well as activities they would like to engage in more frequently. In line with BA guidelines [10], the therapist guided the selection of activities by helping participants to reflect on their personal goals and values and, in turn, on the behaviors and activities aligned with such goals and values to be targeted by the intervention. Each activity represented the content of one session. During this first meeting, participants also received a brief psychoeducation session regarding the theoretical framework underlying BA interventions.

The use of VR was included in the protocol to provide participants with a virtual spatial reference in which to place themselves while visualizing and virtually experiencing the chosen activities. With this aim in mind, we used the Google Earth VR application for Oculus Rift, which allows participants to travel to the place associated with a selected activity and experience the sense of “being there” from a first-person perspective. Participants were asked to identify a specific place associated with the activity to be performed in VR (e.g., a specific place to have a walk; a specific pub to have a drink with some friends). The virtual scenario was entered by means of a head-mounted display (Oculus Rift DK2; Menlo Park, CA, USA) connected to a laptop (Alienware 17 R5 with NVIDIA GTX1070 graphics card and Intel i7 CPU; Miami, FL, USA). The setup also included two sensors and two hand-controllers, which enabled participants to move and explore the environment.

Each session was structured as follows. First, the participants were invited to wear the head-mounted display to start the VR experience. They were guided through the experience by means of a narrative (see Appendix C), which was inspired and adapted from a previous study on the use of mental imagery to envision and plan positive activities [17]. Consistent with Renner et al. (2019), the narrative was structured in order to address the following aspects: (1) focus the attention on the virtual environment to enhance the sense of presence, (2) virtually engage in the planned activity, and (3) concentrate on the positive outcomes of the activity. Subsequently, participants were invited to verbalize their thoughts and emotions in order to develop a solid rationale that justified the importance of planning the activity. Participants were, for instance, asked to recall pleasant memories associated with the activity in the past, identify possible beneficial outcomes associated with the activity or explore the potential barriers and solutions to reintroduce the activity in their lives. Following the principles of BA interventions [10], participants were finally asked to schedule the selected activity following a plan. To do so, each participant received a weekly planner to identify the steps, times and places needed to structure the activity. The same procedure was followed in each session. During the first (i.e., baseline) and last sessions (i.e., end of the treatment), participants were also invited to revise their daily diary. A graphical feedback depicting daily mood and activity level fluctuations was provided to show the connection and reciprocal influence between behavioral patterns and mood shifts. Consistently, participants were invited to reflect on the beneficial effects of engaging in pleasurable activities (or not engaging in any type of activity) on their general mood. 

### 2.5. Measures

Behavioral outcomes: Using a Visual Analogue Scale (VAS) ranging from 0 (not at all) to 100 (extremely), participants were asked to rate their daily activity level (“To what extent have you engaged in activities and be activated today?”) and the rate of savoring during the performed activities (“Today, I have been able to savor and take the most of the things I have done”). In addition, the short form [51] of the Behavioral Activation for Depression Scale (BADS-short) [52] was administrated to monitor the overall level of patients’ daily behavioral activation. The short form of the BADS includes nine items, whose scores are summed to calculate two subscales: behavioral activation and avoidance. Overall, this scale has shown good internal consistency, reliability, construct validity, and predictive validity [51]. For the aim of this study, only the behavioral activation subscale was administered.

Affective outcomes: Similar to previous studies [53], participants were asked to rate their daily mood on VAS ranging from 0 (happy) to 100 (sad) (“Today, I have felt…”). Furthermore, daily depressive symptoms were assessed with the Patient Health Questionnaire–2 (PHQ-2), a two-item self-report measure of depression that has obtained good psychometric properties (i.e., construct and criterion validity) in past research [54]. Scores for each item can range between 0 (not at all) and 3 (extremely). A previous study has shown the suitability of the PHQ-2 for monitoring depressive symptoms using an electronic diary in daily life [55,56].

Specific activity-related outcomes: For each of the four activities included in the protocol, participants were asked to rate their level of motivation on a VAS ranging from 0 (not at all) to 100 (extremely) (“To what extent do you feel motivated to perform this activity?”). In the daily assessment, participants were also asked to report the time spent (i.e., minutes) practicing or planning each activity (e.g., structuring the activity, looking for information, sharing the plan with some friends, etc.).

### 2.6. Procedure

Participants were recruited at the Jaume I University through poster advertisements offering a VR-based brief training for depression. The individuals interested in the study were sent a web-based survey to complete the PHQ-9 and the PANAS. If inclusion criteria were met, participants were invited to attend the laboratory to further explore the exclusion criteria through the administration of the MINI. If all the criteria were satisfied, participants were asked to sign the informed consent and invited to join the study. During this first face-to-face session, participants were also guided through the selection of the personalized activities to be included in the protocol (see Appendix B). The participants were randomly allocated to one of the three baselines using an online randomizing tool.

The participants were provided with an email to get in contact with one of the researchers of the team in case of technical problems or doubts about the protocol. When missing data on two consecutive daily assessments were detected, the participants were contacted and were reminded about the importance of completing the diary. All the participants attended the four BA sessions, so no dropout was observed during the study.

At the end of the study, the participants were invited to attend the laboratory for a final debriefing session.

### 2.7. Data Analysis

The present study is a SCED with multiple baselines across participants (i.e., seven participants allocated to three different baseline lengths) and across behaviors (i.e., four sessions for each participant, one for each of the four activities).

To test the hypotheses related to the behavioral and affective outcomes, we performed several data overlap methods that have been widely used in the SCED literature [57], including the percent of data points exceeding the median (PEM) and the percent of all non-overlapping data (PAND), as well as the more modern non-overlap of all pairs (NAP). These analyses were conducted across participants and all of them included a comparison of baseline-to-post-treatment changes in daily measures. However, while the NAP compares each single point of the baseline phase (A) to each assessment point of the treatment phase (B), the PEM and the PAND are simpler indices that use only part of the data. Particularly, the PEM is calculated with all the points in the treatment phase but only a single point (i.e., the median) in the baseline phase, while the PAND is calculated with the data that remains after removing the minimum number of points that would eliminate all overlap between data from phases A and B [54,55]. All these non-overlap analyses allow calculation of a percentage of non-overlap (i.e., improvement) that can range from 0 to 100. However, emphasis will be made on NAP indices because they generally outperform the remaining indices [57,58], and only this index will be provided for secondary outcomes for readability reasons. In relation to the NAP scores, the median non-overlap of past studies has been proposed as a good comparison measure for interpretation [59,60]. NAP scores higher than 96% reflect very large intervention effects, while indices between 66% and 96% should be interpreted as moderate-to-large effects. For the PAND, moderate-to-large effects would be similar to the NAP (i.e., between 64% and 86%), while non-overlap cut-offs in the PEM are generally stricter and should be over 70% and as close to 100% as possible [59,61,62]. A/B outcome graphs for the visual inspection of the results have been included in the Appendix A.

To test our hypotheses regarding the specific effect of the intervention on each of the four chosen activities, NAP analyses across behaviors were performed. More specifically, each session was considered as a replication of the intervention targeting a different behavior. In other words, each participant had four different baselines, representing the four activities selected at the beginning of the study. For each participant, NAP analyses for each of the four activities were performed by comparing baseline-to-post-treatment changes of the specific activity-related daily measures.

## 3. Results

### 3.1. Behavioral and Affective Outcomes

First, we performed analyses to explore whether the intervention produced a significant change in terms of both behavioral and affective measures. As shown in Table 2, all the participants showed a moderate-to-large improvement in at least one of the two behavioral measures (i.e., activity level and behavioral activation as assessed with the BADS-BA), irrespective of the overlap index used. According to the NAP, five participants improved on both outcomes, while two participants only improved on one of the two measures. Similar results were obtained with the PEM analyses, whereas the PAND index indicated a significant improvement on both variables in all the participants. Moreover, most participants (i.e., five out of seven) also reported increased daily savoring levels based on NAP and PEM analyses, which was generalized to all the participants when considering the PAND index. Overall, effect sizes and the corresponding interpretation were consistent across indices, except for participant 4, who presented a poor response to the treatment on behavioral activation and savoring according to the NAP and the PEM, and a good response based on the PAND.

Regarding the affective measures (see Table 3), the NAP and PEM analyses indicates that six out of seven participants showed a moderate-to-large improvement in daily mood, while all participants significantly increased daily mood levels according to the PAND index. On the other hand, six out of seven participants reported a significant reduction in daily depressive symptoms according to the PAND analyses. The number of participants showing a significant clinical gain was lower when performing NAP (i.e., five out of seven participants) and PEM analyses (i.e., four out of seven participants).

### 3.2. Specific Activity-Related Outcomes

We therefore explored whether the intervention produced a significant change in the variables assessing each of the activities included in the protocol (Table 4).

All participants showed a moderate-to-large improvement in the time spent performing or planning at least one of the four activities. More specifically, five out of seven participants significantly increased the daily time spent performing one or more of the selected activities. One participant reported a significant increase in the time spent practicing three of the four activities, two participants significantly increased the time spent practicing two activities, and one participant showed a significant increase in the time spent practicing one activity. Importantly, the two participants who did not report a significant increase in the time spent practicing the chosen activities (participant 1 and participant 4) did show a large-to-moderate improvement in the time spent planning four and two activities, respectively.

Regarding motivation, only four out of seven participants reported increased rates of motivation to engage in one or more of the activities planned during the intervention.

## 4. Discussion

So far, a growing body of research has shown the efficacy of BA for the treatment of depression. In the present study, we used the Google Earth VR application to introduce the use of VR into a brief BA-based protocol and provide participants with a virtual spatial reference in which to visualize and experience four personalized activities to be re-introduced in their daily routines. 

All the participants showed a significant clinical gain on at least one of the two variables assessing daily activation level, thus suggesting the short-term effectiveness of the protocol to behaviorally activate the participants. Furthermore, most participants reported increased daily rates of savoring. In other words, the VR intervention was not only associated with an adaptive behavioral change, but also with a significant improvement in the quality and enjoyment of the daily activities in more than half participants. Importantly, these findings and those of the remaining outcomes were generally consistent regardless to the overlap index used, with the exception of one participant (i.e., participant 4). These behavioral findings are further strengthened by the secondary analyses designed to explore the behavioral changes in the specific activities included in each participant’s protocol. Indeed, all participants reported a moderate-to-large improvement in the time spent planning and/or practicing one or more activities scheduled during the intervention. The fact that not all the participants significantly increased the engagement in at least one activity should not be considered as a negative outcome. Indeed, the goal of BA interventions is not to abruptly re-introduce an activity into a patient’s daily routine. Rather, to gradually involve the patient in the steps needed to engage in a specific behavior, following his/her personal needs and level of confidence [10]. Considering the brief duration of our protocol, a significant increase in the time spent planning, but not practicing, an activity should be interpreted as an important clinical gain.

Besides, significant improvements in terms of mood and depressive symptoms were also observed in most patients taking part in the study, and results were generally coherent irrespective to the index used. Importantly, one participant (participant 4) did not show any significant gain in any of the affective measures according to the NAP and PEM analyses. This might be explained by the fact that this participant had the lowest average baseline daily mood (M = 29, SD = 14.99) and the lowest average baseline PHQ2 score (M = 0.89; SD = 0.78) from the whole sample (daily mood: M = 48.68, SD = 16.26; PHQ2: M = 2.43, SD = 0.96), thus being approximately 1 SD below the sample average scores. In other words, his baseline was defined by a relatively high average daily mood and very low average depressive symptoms. Accordingly, there might have been less room for clinical improvement, consistent with past research [63]. It is important to note that this finding was observed thanks to the use of a modern overlap index, the NAP, because the analyses with the PAND, a simpler but more likely to be biased method [57], wrongly supported the effectiveness of the intervention for this participant (which was not confirmed by the NAP and the PEM). 

In addition to the findings on mood and depression for patient 4, we also did not observe the expected results in terms of motivation. This might be explained by the design of the study. It might be the case that, although behavioral activation did occur after the administration of the treatment, the enhancement of the patients’ motivation would have required more time and practice. A more intense training over a longer period of time might have been more effective to also increase patients’ motivation.

Overall, the encouraging findings of the present study suggest that our VR-based BA intervention might represent an effective tool to target depressive symptoms, at least in the short term. Unlike traditional BA treatments, the main novelty relies on the use of VR. First, the use of VR has been shown to increase patients’ engagement and motivation towards a treatment which, in turn, might enhance its effectiveness and reduce dropout rates [64,65]. Even though we did not collect any acceptability data regarding the proposed intervention, no dropout was observed throughout the study. Second, VR has been found to lead to the creation of more vivid mental simulations and more intense hedonic expectations as compared to other methods, thus increasing people’s likelihood to seek out those situations in real-life [36]. Our results seem to confirm this hypothesis, since all the participants significantly increased the time spent planning and/or practicing the activities included in the protocol. Third, the technology used in the present study (i.e., Google Earth VR for Oculus Rift) is easily accessible and does not require high economical investment, which makes our VA-based protocol an adequate candidate to be introduced in routine clinical practice [27]. In addition, the proposed intervention allows personalization of the protocol according to the patient’s needs, thus leading to a highly tailored intervention. Finally, these preliminary results may suggest that just four sessions provided over a brief period of time could be effective to alleviate depressive symptoms and behaviorally activate patients, which may be the first step of a psychological treatment targeting depression. In this sense, future research is needed to investigate whether a more intensive training over a longer period of time might further strengthen the effectiveness of the present protocol and increase the patients’ motivation to engage in more pleasurable and adaptive behaviors. Moreover, future research should explore whether the present protocol could be introduced as a component of a broader intervention for MDD patients.

The present study is not free of limitations. First, the sample was mainly composed of female undergraduate students. Future studies are needed to replicate the present findings in a more heterogeneous sample. Additionally, although the use of VR can represent an added value, the specific use of the Google Earth VR application might have limited the possibilities offered by our intervention. Despite the high level of personalization, patients were asked to select their personally significant activities from a predetermined list, which was created based on previous studies and, most importantly, on the feasibility to virtually visualize the activity through the Google Earth VR application (i.e., outdoor activities). Third, our sample was mainly composed of individuals suffering from moderate to moderate-severe depressive symptoms, and whether our protocol could be similarly effective in patients with more severe symptoms should be further explored. Finally, further studies are needed to disentangle the role of VR in a BA intervention. This study does not allow conclusion of whether the use of VR as compared to visual imagery to anticipate a future positive activity can actually increase the motivation and likelihood to engage in that situation. In this sense, future studies are needed that focus on the specific added value of VR-based BA protocol as compared to a traditional BA intervention, as well as studies that replicate our findings using a larger number of participants and a more robust experimental design.

## Figures and Tables

**Table 1 jcm-11-01262-t001:** Characteristics of the recruited sample.

ID	Age	Sex	Medication	Group	PHQ-9	PANAS-PA
1	23	f	No	3	15	15
2	21	f	No	1	21	21
3	20	f	No	3	14	14
4	23	m	Yes	2	14	19
5	26	f	No	3	14	18
7	23	f	Yes	1	15	23
8	20	f	No	2	18	17

Group 1: 8 days of baseline (phase A), 23 days of treatment monitoring (phase B); Group 2: 9 days of baseline (phase A), 22 days of treatment monitoring (phase B); Group 3: 12 days of baseline (phase A) and 19 days of treatment monitoring (phase B). (PHQ-9 = Patient Health Questionnaire—9 items; PANAS-PA: Positive and Negative Affect Schedule—Positive Affect subscale).

**Table 2 jcm-11-01262-t002:** Results of the NAP, PAND and PEM analyses across-participants in relation to the behavioral outcome variables.

ID	Activity Level			BADS-BA			Savoring		
	NAP	PAND	PEM	NAP	PAND	PEM	NAP	PAND	PEM
1	80 *	80 *	87 *	74 *	72 *	67	72 *	76 *	73 *
2	76 *	71 *	79 *	76 *	76 *	75 *	72 *	71 *	75 *
3	60	67 *	64	66 *	67 *	64	61	67 *	64
4	87 *	86 *	95 *	53	77 *	30	40	70 *	33
5	80 *	82 *	100 *	83 *	79 *	85 *	85 *	79 *	82 *
7	66 *	67 *	64	75 *	76 *	86 *	78 *	76 *	71 *
8	78 *	89 *	100 *	83 *	82 *	86 *	74 *	86 *	95 *
	6/7	7/7	5/7	6/7	7/7	4/7	5/7	7/7	5/7

* = NAP indices over 66% (moderate-to-large effect); * = PAND indices over 64% (moderate-to-large effect); * = PEM indices over 70 (moderate-to-large effect). BADS-BA: Behavioral Activation for Depression Scale-Behavioral Activation subscale; NAP: non-overlap of all pairs; PAND: percentage of all non-overlapping data; PEM: percentage of data exceeding the median.

**Table 3 jcm-11-01262-t003:** Results of the NAP, PAND and PEM analyses across-participants in relation to the affective outcome variables.

ID	Daily Mood			PHQ2		
	NAP	PAND	PEM	NAP	PAND	PEM
1	70 *	76 *	87 *	79 *	76 *	73 *
2	89 *	86 *	93 *	49	64	68
3	67 *	71 *	75 *	66 *	71 *	43
4	42	70 *	33	43	65 *	50
5	67 *	75 *	76 *	78 *	79 *	94 *
7	88 *	81 *	93 *	79 *	81 *	93 *
8	79 *	83 *	90 *	79 *	79 *	86 *
	6/7	7/7	6/7	5/7	6/7	4/7

* = NAP indices over 66% (moderate-to-large effect); * = PAND indices over 64% (moderate-to-large effect); * = PEM indices over 70 (moderate-to-large effect). PHQ2: Patient Health Questionnaire—2; NAP: non-overlap of all pairs; PAND: percentage of all non-overlapping data; PEM: percentage of data exceeding the median.

**Table 4 jcm-11-01262-t004:** Results of the NAP analyses across-behaviors in relation to the specific activity-related outcome variables.

	Motivation				Time Spent				Time Planning			
ID	Act. 1	Act. 2	Act. 3	Act. 4	Act. 1	Act. 2	Act. 3	Act. 4	Act. 1	Act. 2	Act. 3	Act. 4
1	78 *	79 *	90 *	90 *	57	64	56	60	73 *	67 *	74 *	79 *
2	39	74 *	36	41	93 *	75 *	58	76 *	67 *	60	64	73 *
3	66 *	95 *	81 *	65	53	49	84 *	63	31	44	81 *	58
4	87 *	100 *	99 *	91 *	47	50	50	46	53	66 *	80 *	39
5	65	64	45	20	71 *	74 *	62	42	62	65	65	58
7	17	22	45	36	67 *	56	23	56	10	27	32	47
8	56	34	45	60	96 *	66 *	56	54	87 *	45	52	40

* = NAP indices over 66% (moderate-to-large effect). Act: Activity.

## Data Availability

The data presented in this study are available on request from the corresponding author.

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
