# Peer review of "Behavioral Activation through Virtual Reality for Depression: A Single Case Experimental Design with Multiple Baselines"

_jcm, 2022, doi:10.3390/jcm11051262_

Round 1

Reviewer 1 Report

The issue of how to make BA work is clinically really useful and so this study is a potentially a useful addition.  The integration of VR and BA has not been extensively studied.
My comments are as follows:
Introduction:
The review of the effectiveness and efficacy of BA is too slight - there is much more evidence across a number of meta analyses and these should be referenced. 
The study uses the terms efficacy at times - this is a multiple baseline study and not a fully powered RCT and therefore it is inappropriate to use the term efficacy.  Please use effectiveness throughout. 
Method
Include the ethics review reference and committee earlier. 
It is inappropriate in the method to argue to case for the SCED - just report what you did and how you did it to a level that would ensure replication by other researchers. 
The inclusion and exclusion criteria and also the means of assessment means that it is not appropriate to talk about MDD - this needs to be toned down more to say participants with elevated depressive symptoms.  Would did the MINI and what was their other involvement in the study?  Would did the assessments and were they independent of the allocation of participants.  How were participants allocated to different baselines (which programme) and was this completed by a team separate to the study?  Was anything blinded at all?        
Add a medication column to Table 1.  
Supply the psychometric foundations of the BADS and PHQ-2.
A reference for the interpretation of the NAP is required. 
Other non-overlap statistics are available and it is the norm to also use and report PEM and PAND to give a better overview.  Add these.        
Graphing of time series data is the norm for SCED and is used to contextualise non-overlap results.  Each participant needs an A/B outcome graph on the same one behavioural and affective outcome measure. 
The discussion cold be shortened and sharpened.           
As a side note: was there any acceptability data collected?  It would be good to report this.  The number of treatment sessions attended could be worked out and reported - and also the dropout rate reported clearly.    

Author Response

The review of the effectiveness and efficacy of BA is too slight - there is much more evidence across a number of meta analyses and these should be referenced. 

We thank the reviewer for this suggestion. We completely agree that further literature regarding the effectiveness and efficacy of BA should be referenced in the manuscript. We have now revised the introduction in order to provide a stronger theoretical background supporting the importance of BA treatments (from line 65; line 115).

The study uses the terms efficacy at times - this is a multiple baseline study and not a fully powered RCT and therefore it is inappropriate to use the term efficacy.  Please use effectiveness throughout. 

Following the reviewer’s suggestion, we have replaced the word “efficacy” with “effectiveness” when referring the the objective and findings of our study.

Include the ethics review reference and committee earlier. 

We have now incuded the statement in the section “2.1 study design” (from line 143).

It is inappropriate in the method to argue to case for the SCED - just report what you did and how you did it to a level that would ensure replication by other researchers.

Thank you for the suggestion. We have now deleted the paragraph which justified the selection of the study design, so that the “2.1 study design” section specifically focuses on the design and structure of our study.

The inclusion and exclusion criteria and also the means of assessment means that it is not appropriate to talk about MDD - this needs to be toned down more to say participants with elevated depressive symptoms.  

The reviewer is right when he/she argues that it is not appropriate to talk about MDD, as this was not an inclusion criteria of our study. All participants, however, met the criteria for a MDD as assessed with the MINI, which was adminstrated to check for the exclusion criteria. We have now toned down our results, emphasizing that, even though all participants met the inclusion crtieria for a MDD, this was not an inclusion criteria (and we also note that the baseline PHQ9 also indicated moderate levels of depression in most participants) (from line 161; from line 440 ).

Would did the MINI and what was their other involvement in the study?  Would did the assessments and were they independent of the allocation of participants.  How were participants allocated to different baselines (which programme) and was this completed by a team separate to the study?  Was anything blinded at all?

The baseline assessement (i.e., the MINI) was conducted by two authors of this article (D.C., I.O.B.), who were indipendent from the allocation of participants to each baseline. Allocation was blinded and performed using an online randomizing tool (www.randomizer.org) (line 260).

Add a medication column to Table 1.

We have added a medication column to Table 1.

Supply the psychometric foundations of the BADS and PHQ-2.

We have included the reference showing that the short form of the BADS has obtained good internal consistency, reliability, construct validity, and predictive validity in past research (Manos et al., 2011). Furthemore, we have underlined the good psychometric proporties of the PHQ2 (Kroenke et al. 2002), as well as its suitability to be used with daily electronic diaries for the monitoring of depressive symptoms (Bauer et al., 2018) (from line 234).

A reference for the interpretation of the NAP is required.

We have included the required references.

Other non-overlap statistics are available and it is the norm to also use and report PEM and PAND to give a better overview.  Add these.

We have run new analyses to calculate both PAND and PEM indixes for the primary outcome measures (behavioral and affective outcomes), which have been included in the methods (from line 273), results and discussion.

Graphing of time series data is the norm for SCED and is used to contextualise non-overlap results.  Each participant needs an A/B outcome graph on the same one behavioural and affective outcome measure. 

We thank the reviewer for the suggestions. We now provide the visual inspection analyses of the primary outcome variables (daily mood, PHQ2, savoring, daily activity, BADS-BA) as part of the supplementary material (line 294).

The discussion cold be shortened and sharpened.

Thank you for the suggestion. The discussion has been revised and redundant information has been deleted.

As a side note: was there any acceptability data collected?  It would be good to report this.  

Unfortunately, no acceptability data was collected for the present study. We have included this information as a limitation of the study at the end of the discussion (line 413).

The number of treatment sessions attended could be worked out and reported - and also the dropout rate reported clearly.  

All the participants attended the four BA sessions, so no dropout was observed during the study. We have now included this information in the “2.6 procedure” section (line 265; line 413).

Reviewer 2 Report

Overall, a very interesting paper with important implications for an emerging field, such as the use of virtual reality for the treatment of emotional disorders. The experimental design is presented correctly and the presentation of results is consistent with the design.

In my opinion, in this type of experimental designs with multiple baselines, it is convenient to present the visual inspection analysis using graphs of clinical changes. Likewise, it is also recommended to use analysis of clinical significance through case-by-case analysis, however, the analyzes used are adequate.

For future studies, it would be good to analyze the proposed protocol with a larger number of participants and a more robust experimental design.

Author Response

In my opinion, in this type of experimental designs with multiple baselines, it is convenient to present the visual inspection analysis using graphs of clinical changes. Likewise, it is also recommended to use analysis of clinical significance through case-by-case analysis, however, the analyzes used are adequate.

We thank the reviewer for the suggestions. We now provide the visual inspection analyses of the primary outcome variables (daily mood, PHQ2, savoring, daily activity, BADS-BA) as part of the supplementary material (line 294). We have run new analyses to calculate both PAND and PEM indixes for the primary outcome measures (behavioral and affective outcomes), which have been included in the methods (from line 273), results and discussion.

For future studies, it would be good to analyze the proposed protocol with a larger number of participants and a more robust experimental design.

We completely agree with the reviewer. We have included these two points at the end of the discussion (line 447).

This manuscript is a resubmission of an earlier submission. The following is a list of the peer review reports and author responses from that submission.